# VIDEODiT: BRIDGING IMAGE DIFFUSION TRANS-FORMERS FOR STREAMLINED VIDEO GENERATION

## ABSTRACT

We present VideoDiT, a streamlined video generation framework adapted from pre-trained image generation models. Unlike previous methods that simply add temporal layers to image diffusion models, we enhance both the tokenizer, implemented with the variational autoencoder (VAE), and the diffusion model. We emphasize the importance of combining 3D VAE compression with knowledge from pre-trained image diffusion models to achieve efficient video generation, though the tight coupling between image diffusion models and 2D VAEs poses significant challenges. To address this, we introduce the Distribution-Preserving VAE (DP-VAE), which encodes key frames in a video clip using the original 2D VAE while compressing non-key frames with a 3D VAE for spatiotemporal modeling. A regularization term ensures alignment between the 3D video latent space and the 2D image latent space, facilitating seamless transfer of pre-trained diffusion models. Leveraging the Diffusion Image Transformers (DiT) architecture and incorporating 3D positional embeddings, we extend 2D attention into 3D with negligible increased parameters. Furthermore, leveraging our proposed DP-VAE, VideoDiT supports joint image-video training, preserving the spatial modeling capabilities of the base model while excelling in both image and video generation. Extensive experiments validate the effectiveness of our approach.

## 1 INTRODUCTION

In recent years, generative models for images and videos have achieved significant milestones. While image generation models like Stable Diffusion 3 (SD3) (Esser et al., 2024), DALL·E 3 (Betker et al., 2023), and the Flux series (Flux) have reached maturity and are poised for real-world applications, the progress in video generation has been less remarkable. Since the release of the SORA (Brooks et al., 2024) model, advancements in this space have been limited, and there remains a lack of open-source models capable of producing high-quality video clips. The key challenge in video generation lies in efficiently modeling spatiotemporal dynamics intrinsic to video data.

Modern image generation relies on two foundational components to model spatial dynamics: a variational autoencoder (VAE) (Kingma, 2013) for compressing visual signals into compact latent representations, and denoising diffusion models (Ho et al., 2020; Dhariwal & Nichol, 2021), which offer a robust framework for modeling arbitrary distributions in this latent space. This VAE+Diffusion framework has proven highly effective, particularly for text-to-image (T2I) generation tasks (Rombach et al., 2022), and it has been widely adopted for video generation as well. However, two distinct approaches have emerged in adapting this framework for videos.

The first approach extends pre-trained T2I models into text-to-video (T2V) models by introducing temporal layers to the diffusion model while keeping the image VAE unchanged (Blattmann et al., 2023b; Guo et al., 2024; Wang et al., 2024; Girdhar et al., 2023; Blattmann et al., 2023a). These methods leverage the strong spatial modeling abilities of T2I models, which are derived from training on billions of images. However, since pre-trained T2I models are tightly coupled with their corresponding VAEs, these approaches require the video diffusion model to operate within a 2D latent space. This setup places the entire burden of temporal dynamics modeling on the diffusion process, leading to an imbalance in the representation of spatial and temporal information, ultimately resulting in suboptimal performance and efficiency.

The second approach, as proposed in recent works (Gupta et al., 2023; Brooks et al., 2024), advocates for designing T2V models based on 3D spatiotemporal VAEs paired with diffusion models that operate in 3D latent spaces. While this method is more efficient in compressing video data, the intrinsic link between diffusion models and their corresponding latent spaces poses a challenge. Since diffusion models are trained to capture specific latent distributions, transitioning from a 2D to a 3D latent space necessitates training video diffusion models from scratch. This leads to a significant loss of prior knowledge embedded in pre-trained T2I models and demands substantial computational and data resources.

In this work, we present **VideoDiT**, a novel framework that aims to streamline video generation by leveraging a well-trained T2I model. The primary challenge is maximizing the benefits of these pre-trained models without compromising the efficiency of the video generation process. Our key innovation is the **Distribution-Preserving VAE (DP-VAE)**, which combines the compression efficiency of 3D VAEs with the latent space alignment of 2D VAEs. DP-VAE processes video sequences as a series of Groups of Pictures (GoPs), where key frames are encoded using the original 2D VAE, and non-key frames undergo spatiotemporal compression via a 3D VAE. This method ensures that the resulting residual representations maintain the same dimensionality as those of the key frames, allowing seamless integration into the pre-trained diffusion framework. By further incorporating a regularization term, we align the 3D and 2D latent distributions, enabling the pre-trained image diffusion model to be efficiently transferred to the video generation task.

Regarding the diffusion model, our architecture is built on diffusion image transformers (DiT), which have emerged as a promising alternative to previous U-Net-based designs (Ronneberger et al., 2015; Ho et al., 2020; Dhariwal & Nichol, 2021), due to their simpler structure and enhanced scalability (Peebles & Xie, 2023; Esser et al., 2024). Notably, we find that directly applying the original 2D spatial attention weights to 3D global attention produces consistent and stable video output without introducing additional parameters. By incorporating 3D positional embeddings during the patchifying process, we initialize the video diffusion model from a strong base. To preserve the spatial modeling capabilities of the text-to-image (T2I) model, we further utilize DP-VAE to introduce joint image-video training. This enables VideoDiT to excel in both image and video generation tasks. Extensive experiments validate the effectiveness of our method in generating high-quality images and videos. Notably, our method achieves comparable or even superior performance while utilizing only approximately 10% of the data employed by existing methods.

## 2 RELATED WORK

### 2.1 VARIATIONAL AUTOENCODER

Variational Autoencoders (VAEs) (Kingma, 2013) is originally proposed as generative models, optimized by maximizing the Evidence Lower Bound (ELBO). Previously, VQ-VAEs (Van Den Oord et al., 2017; Razavi et al., 2019; Esser et al., 2021) compress videos into discrete tokens for categorical generative modeling. Recently, VAEs have been commonly utilized as codecs to bridge original visual signals with continuous latent variables in text-to-image latent diffusion models (Rombach et al., 2022; Pernias et al.; Esser et al., 2024). Specifically, VAE is used to compress high-dimensional visual data, thereby reducing spatial dimensions to decrease computational complexity. More importantly, by constraining the latent space size and balancing signal fidelity with perceptual realism in the loss function, the VAE encoder effectively discards high-frequency, textural information while retaining predominantly semantic information. This reduction in data distribution complexity facilitates the modeling tasks of diffusion models. Text-to-video generation models (Gupta et al., 2023; Kondratyuk et al., 2024; Brooks et al., 2024) usually utilize 3D VAE that compresses videos in both spatial and temporal dimensions. Temporally causal 3D VAE (Yu et al., 2024) is proposed to support both image and video generation. However, retraining a 3D VAE often results in an entirely new latent distribution, making it challenging to leverage the original well-trained T2I model's conceptual understanding and generation capabilities. Consequently, these methods typically require substantial computational resources to train the video diffusion models entirely from scratch. In contrast, our proposed DP-VAE effectively compresses the video data along both temporal and spatial dimensions while maintaining the original distribution. This enables a seamless transition of the T2I model into a T2V model.

## 2.2 TEXT-TO-VIDEO DIFFUSION MODELS

Diffusion models (Ho et al., 2020; Dhariwal & Nichol, 2021) have proven effective in modeling high-dimensional perceptual data (Ramesh et al., 2021; 2022; Saharia et al., 2022). Building on the success of text-to-image generation, text-to-video generation has garnered increasing attention, given the significant role of video multimedia in daily life. Previous video diffusion models (Singer et al., 2022; Ho et al., 2022; Blattmann et al., 2023b; Wang et al., 2024) often extend pre-trained T2I models and build up hierarchical cascaded pipelines that include keyframe generation, temporal up-sampling, and spatial upsampling. This approach results in substantial costs for training, inference, and deployment. Worsely, during the keyframe training phase, the direct downsampling of video data along both temporal and spatial dimensions results in significant information loss, which substantially impairs the modeling of realistic natural distributions and physical phenomena. Recently, advanced techniques (Gupta et al., 2023; Kondratyuk et al., 2024; Brooks et al., 2024) have proposed using 3D VAEs to compress videos into latent variables, reducing dimensions in both spatial and temporal domains. This enables more efficient diffusion model training and improves model capabilities. On the other side, traditionally, U-Net-based (Ronneberger et al., 2015) architectures have dominated diffusion models due to their multi-scale modeling capabilities and skip connections that preserve fine-grained information. However, transformer-based image diffusion models (Peebles & Xie, 2023; Esser et al., 2024; Brooks et al., 2024) have recently emerged as promising alternatives, demonstrating robust performance and scalability. This shift is attributed to their simpler architecture and enhanced scalability compared to U-Net structures. In this paper, we develop a framework for efficiently bridging a well-trained T2I diffusion transformer towards a video generator.

## 3 METHOD

Our proposed VideoDiT contains two core contributions in terms of the fundamental components of latent diffusion models (Rombach et al., 2022; Esser et al., 2024), namely, the VAE and the generation model. In Section 3.1, we introduce our proposed Distribution-Preserving VAE (DP-VAE), which can compress both image and video data while maintaining the original distribution. This preservation facilitates the seamless adaptation of subsequent diffusion model training. In Section 3.2, we propose an efficient method to convert an image diffusion transformer into a video version, requiring negligible additional parameters.

### 3.1 DISTRIBUTION-PRESERVING VAE

In latent diffusion models, VAEs play a pivotal role in dimensionality reduction and the elimination of irrelevant information. We introduce the **Distribution-Preserving VAE (DP-VAE)**, which integrates compressed residual temporal information into the original 2D latent space and applies regularization to the latent variables. This approach effectively transforms a 2D VAE into a 3D VAE while maintaining the original distribution, thereby facilitating the seamless adaptation of pre-trained T2I models for video generation.

Inspired by video compression techniques (Lu et al., 2019; Hu et al., 2021), we partition the original video signal into multiple Groups of Pictures (GoPs), each comprising one key frame and several non-key frames sampled every $t_g$ frames. For each GoP, the key frame, $\boldsymbol{x}_k$, is encoded using the pre-trained 2D VAE encoder, $\mathcal{E}_k$, producing a latent representation, $\boldsymbol{z}_k$. Concurrently, the entire video segment $\boldsymbol{x}$ is processed by a 3D VAE encoder $\mathcal{E}_r$, which performs spatiotemporal redundancy removal and compresses the left information of non-key frames into the residual latent representation $\boldsymbol{z}_r$, consistent in dimensionality with $\boldsymbol{z}_k$. This compression includes downsampling in both spatial and temporal dimensions, with temporal downsampling applied after each of the first two spatial downsampling operations, resulting a temporal downsampling factor of $t_g = 4$. The combined latent variable, $\boldsymbol{z}$, is obtained by adding $\boldsymbol{z}_k$ and $\boldsymbol{z}_r$. The detailed architectures of $\mathcal{E}_r$ and the decoder $\mathcal{D}$ are described in the supplementary material. This process is formalized as follows:

$$\boldsymbol{z} = \mathcal{E}_k(\boldsymbol{x}_k) + \mathcal{E}_r(\boldsymbol{x}). \tag{1}$$

On the decoder side, the 3D latent variable $\boldsymbol{z}$, is decoded via 3D decoder $\mathcal{D}$ and gets $\hat{\boldsymbol{x}}$. Then we can calculate the reconstruction loss between original video $\boldsymbol{x}$ and reconstructed video $\hat{\boldsymbol{x}}$ with

Figure 1: **Illustration of the proposed DP-VAE.** The video is compressed by separately encoding key frames and residuals, which are then added to obtain the 3D latent variable $z$. The latent variable $z$ is decoded using the 3D decoder $\mathcal{D}$ to get the reconstruction $\hat{x}$. Additionally, $z$ is decoded through $\mathcal{D}_k$ for regularization.

$$\mathcal{L}_{\text{recon}} = \|x - \hat{x}\|^2. \tag{2}$$

Furthermore, to align the distribution of the 3D latent variable, $z$, with the features extracted by the original VAE encoder and thereby enable seamless adaptation for subsequent diffusion model training, we introduce a distribution-preserving loss as a regularization term. Specifically, this is achieved by combining the L2 loss between the key frames, $x_k$, and their reconstructions, $\hat{x}_k$, decoded via the decoder of the pre-trained 2D VAE, $\mathcal{D}_k$, with the L2 loss between the statistical parameters of the 2D latent variables, $z_k$, and the 3D latent variables, $z$. The process is expressed as

$$\mathcal{L}_{\text{reg}} = \|x_k - \hat{x}_k\|^2 + \|\mu_z - \mu_{z_k}\|^2 + \|\sigma_z - \sigma_{z_k}\|^2, \tag{3}$$

where $\mu_z$ and $\mu_{z_k}$ denote the means of $z$ and $z_k$, respectively, and $\sigma_z$ and $\sigma_{z_k}$ denote their corresponding standard deviations.

Besides, to enhance the perceptual quality, we incorporate a 3D adversarial loss (Isola et al., 2017) and the LPIPS loss (Zhang et al., 2018). The overall training objective is written as

$$\mathcal{L} = \lambda_{\text{recon}}\mathcal{L}_{\text{recon}} + \lambda_{\text{reg}}\mathcal{L}_{\text{reg}} + \lambda_{\text{disc}}\mathcal{L}_{\text{disc}} + \lambda_{\text{LPIPS}}\mathcal{L}_{\text{LPIPS}}, \tag{4}$$

where $\lambda_{\text{recon}}$, $\lambda_{\text{reg}}$, $\lambda_{\text{disc}}$, and $\lambda_{\text{LPIPS}}$ denote the weighting parameters for reconstruction, regularization, discrimination, and perceptual quality, respectively. We set $\lambda_{\text{recon}}$, $\lambda_{\text{reg}}$, $\lambda_{\text{disc}}$, and $\lambda_{\text{LPIPS}}$ to 1, 1, 0.01, and 0.1 by default.

During training, the encoder $\mathcal{E}_k$ and decoder $\mathcal{D}_k$ of key frames are frozen since they are well-trained, while the 3D encoder $\mathcal{E}_r$ and parameters of the 3D decoder $\mathcal{D}$ remain learnable.

## 3.2 UNIFYING IMAGE DIFFUSION TRANSFORMER TO VIDEO GENERATOR

**Spatial to Global Attention.** Adapting a U-Net-based T2I model (Rombach et al., 2022) with convolutions to generating videos (Blattmann et al., 2023b) necessitates the incorporation of additional temporal layers. This adaptation introduces a substantial number of extra parameters, and training the temporal layers from scratch can undermine the knowledge embedded in the original spatial weights. In contrast, Transformers (Vaswani et al., 2017), which are composed of stacked attention blocks which operate on any number of tokens, while the feed-forward layers process each token individually. This architecture facilitates a seamless transition from image to video processing, efficiently modeling both spatial and temporal dependencies without increasing the number of parameters or compromising the integrity of pre-trained spatial features.

As illustrated in Figure 2, for image inputs $z_i \in \mathbb{R}^{h \times w \times d}$, the original spatial attention mechanism first reshapes the input to $(h \times w) \times d$ before passing it through the attention modules. Due to the absence of temporal correlation and interaction between individual images, each generated image

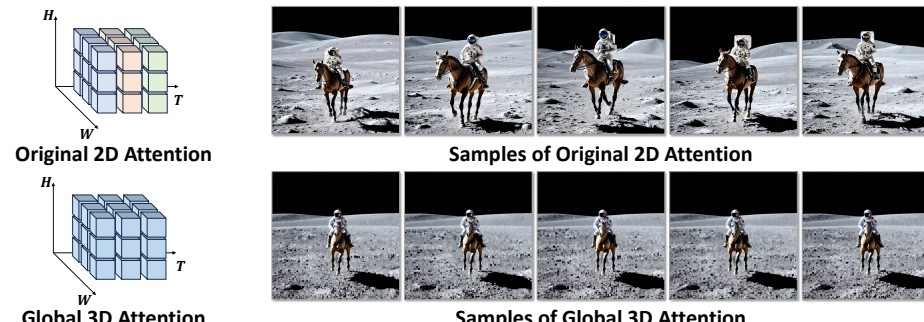

Figure 2: Illustration of 2D and 3D attention mechanisms (left) and their corresponding generated results (right) using a pre-trained image diffusion transformer.

remains entirely distinct. In contrast, for video inputs $z_v \in \mathbb{R}^{t \times h \times w \times d}$, reshaping to $(t \times h \times w) \times d$ and directly applying the attention mechanism of the pre-trained T2I transformer model leads to the generation of "videos" where the main subjects remain static, while block artifacts emerge in the details. This approach achieves partial temporal consistency and serves as an effective initialization strategy without introducing additional parameters.

**3D Patchify.** Although attention operations are applied to all visual tokens, each token incorporates only the original spatial positional information and does not encode its temporal position. Therefore, we incorporate global 3D positional embeddings into the patchify process. Specifically, we introduce an additional branch to conduct patchify operation and add 3D positional embeddings to the flatten input embeddings. The input embeddings are then projected through a zero-convolution (Singer et al., 2022; Zhang et al., 2023b) for smooth updating. Implementation details can be found in the supplemental material.

**Joint Image-Video Training.** Joint training of images and videos not only preserves image generation capabilities but also enhances the learning of comprehensive concepts, leading to superior video generation performance (Gupta et al., 2023). Our proposed DP-VAE inherently supports joint image-video training by ensuring that latent variables for both modalities adhere to the same distribution. During training, a probability parameter, $p$, determines whether each iteration focuses on video generation or image generation. Detailed implementation procedures are provided in the supplementary material.

## 4 EXPERIMENTS

### 4.1 IMPLEMENTATION DETAILS

**DP-VAE.** We employ the open-source Stable Diffusion 3 medium (Esser et al., 2024) as our base model. The training dataset for DP-VAE comprises a self-collected, high-quality video dataset. We train the DP-VAE using the AdamW optimizer (Loshchilov, 2017) for 600K iterations with a batch size of 8 and a learning rate of $5 \times 10^{-5}$. During training, input videos are first resized to ensure that their longer edge does not exceed 1080 pixels. Subsequently, the videos are cropped into clips with a resolution of $256 \times 256$ pixels and sampled to 32 frames. We maintain the original frame rate of approximately 25 frames per second (fps) throughout the training process.

**Video Diffusion Transformer.** The video training dataset comprises 1 million videos randomly sampled from WebVid-10M (Bain et al., 2021) and 300K self-collected videos from Pexels (Pexels, 2024). The training dataset for image generation is JourneyDB (Sun et al., 2024). We first pre-train the diffusion model on the 1 million low-resolution dataset for 200K iterations, with resolution of $320 \times 320$. Then we fine-tune the model on the 300K high-resolution dataset for 200K iterations, with resolution of $512 \times 512$. The frame number is 96 and we keep the original fps as approximately 25. During training, we maintain an exponential moving average (EMA) of model weights over training with a decay of 0.9999. The learning rate is $5 \times 10^{-5}$ and the batch size is 64. Training is conducted with an 80% probability on video generation tasks and a 20% probability on image

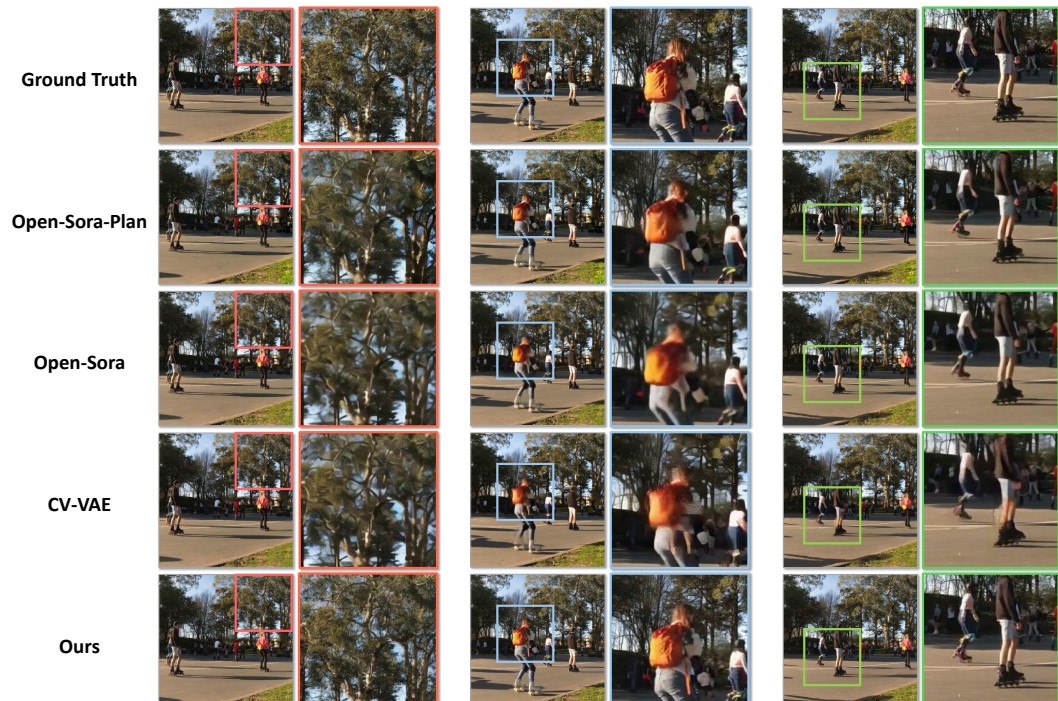

Figure 3: **Qualitative comparison of our proposed DP-VAE and other methods.** While other approaches suffer from varying levels of blur artifacts and texture deterioration, our method excels in both signal fidelity and perceptual quality. Best viewed with a zoomed-in view.

| Method | Parameters | PSNR (↑) | SSIM (↑) | LPIPS (↓) |
|---|---|---|---|---|
| Open-Sora-Plan (Chen et al., 2024b) | 239M | 27.63 | 0.931 | 0.1249 |
| Open-Sora (Zheng et al., 2024) | 393M | 36.19 | 0.962 | 0.1013 |
| CV-VAE (Zhao et al., 2024) | 256M | 33.31 | 0.947 | 0.1031 |
| DP-VAE (Ours) | 195M | **37.53** | **0.980** | **0.0444** |

Table 1: Quantitative results of 3D VAE's reconstruction performances.

generation tasks. The resolutions for both image and video generation training are maintained consistently. Furthermore, we apply a 10% probability of dropping the text prompt for classifier-free guidance (Ho & Salimans, 2021). All experiments are conducted on 8 NVIDIA H100 GPU (80G).

**Evaluation Metrics.** We evaluate the reconstruction performance of our model using three widely adopted metrics: Peak Signal-to-Noise Ratio (PSNR), Structural Similarity Index Measure (SSIM) (Wang et al., 2004), and Learned Perceptual Image Patch Similarity (LPIPS) (Zhang et al., 2018). These metrics are assessed on 50 videos randomly sampled from Pexels (Pexels, 2024), each comprising 64 frames at a resolution of $512 \times 512$ pixels. To evaluate the quality of the generated videos, we employ zero-shot Frechet Video Distance (FVD) (Unterthiner et al., 2018) and Inception Scores (IS) (Salimans et al., 2016) on the UCF101 (Soomro, 2012) dataset.

## 4.2 COMPARISON OF RECONSTRUCTION AND GENERATION

We compare our method against three open-source video VAEs, each employing $8\times$ spatial downsampling and $4\times$ temporal downsampling factors: Open-Sora-Plan VAE (Chen et al., 2024b), Open-Sora VAE (Zheng et al., 2024), and CV-VAE (Zhao et al., 2024). Figure 3 presents a qualitative comparison of reconstruction results among these methods. Unlike other approaches that produce blurred artifacts and inaccurate textures, our method achieves superior signal fidelity and perceptual quality. Furthermore, as shown in Table 1, our method outperforms all others significantly in all quantitative evaluation metrics.

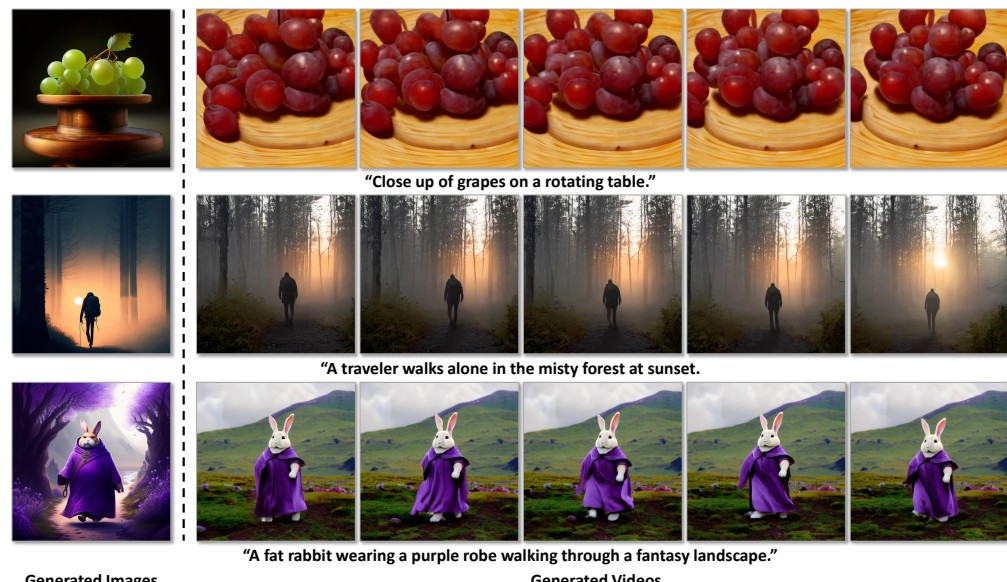

"Close up of grapes on a rotating table."

"A traveler walks alone in the misty forest at sunset.

"A fat rabbit wearing a purple robe walking through a fantasy landscape."

**Generated Images**                                    **Generated Videos**

Figure 4: Illustration of images and videos generated by our proposed VideoDiT.

| Method | Video Training Data | FVD (↓) | IS (↑) |
|---|---|---|---|
| Make-A-Video (Singer et al., 2022) | 20M | 367.20 | 33.00 |
| VideoFactory (Wang et al., 2023a) | 130M | 410.00 | - |
| PYoCo (Ge et al., 2023) | 22.5M | 355.20 | 47.46 |
| Lavie (Wang et al., 2023b) | 20M | 526.30 | - |
| Factorized-Dreamer (Yang et al., 2024) | 19.3M | 503.93 | 33.27 |
| I2V-XL (Zhang et al., 2023c) | 35M | 424.87 | 28.78 |
| CogVideo (Hong et al., 2023) | 10M | 701.60 | 25.27 |
| MagicVideo (Hong et al., 2023) | 10M | 655.00 | - |
| Video LDM (Blattmann et al., 2023b) | 10M | 550.60 | 33.45 |
| AnimateDiff (Guo et al., 2024) | 10M | 598.83 | 35.18 |
| VideoCrafter2 (Chen et al., 2024a) | 10M | 674.09 | 40.28 |
| VideoFusion (Luo et al., 2023) | 10M | 639.90 | 17.49 |
| Show-1 (Zhang et al., 2023a) | 10M | 394.46 | 35.42 |
| VideoDiT (Ours) | (1+0.3) M | 428.52 | 33.04 |

Table 2: Zero-shot results of text-to-video models on UCF101.

Figure 4 presents the results generated by our proposed VideoDiT, which is capable of producing both images and videos. We also assess the zero-shot text-to-video generation performance of our method by comparing it with other methods on the UCF101 video dataset, as illustrated in Table 2. Our proposed method achieves a competitive Frechet Video Distance (FVD) score of **428.52** and an Inception Score (IS) of **33.04**. Notably, our method is trained on a video dataset of only 1.3 million videos, comprising 1 million low-resolution videos and 300K high-quality videos.

### 4.3 ABLATION STUDY

For the ablation study experiments, we utilized a resolution of $256 \times 256$ pixels and processed video sequences comprising 32 frames each. During training, we employed a batch size of $64$ and trained the models for 60K iterations. The training dataset consisted of 1 million videos sampled from WebVid-10M. The probability $p$ of training on video generation tasks was set to 100%, except for the ablation of joint image-video training, where it is set to 80%. All other parameter settings remained consistent with the default configuration outlined in Section 4.1.

**Distribution Preserving 3D VAE (DP-VAE)**. As Figure 5 illustrates, before training the diffusion model, employing a 3D VAE without incorporating a 2D VAE to compress key frames and with-

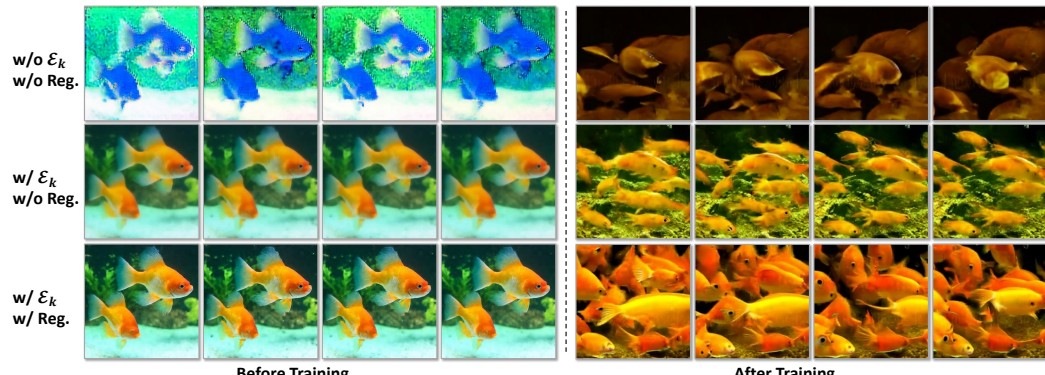

**Before Training**       **After Training**

Figure 5: Qualitative ablation study on our proposed DP-VAE.

| 2D VAE | Regularization | PSNR (↑) | SSIM (↑) | LPIPS (↓) | FVD (↓) | IS (↑) |
|--------|----------------|----------|----------|-----------|---------|--------|
| × | × | 35.57 | 0.9706 | 0.0963 | 683.76 | 23.18 |
| × | ✓ | 33.85 | 0.9643 | 0.1183 | 654.73 | 22.37 |
| ✓ | × | **38.18** | **0.9849** | **0.0380** | 645.28 | 27.56 |
| ✓ | ✓ | 37.53 | 0.9806 | 0.0444 | **556.42** | **29.61** |

Table 3: Quantitative ablation study on our proposed DP-VAE.

out introducing regularization results in entirely erroneous decodings. Introducing a 2D VAE to establish a keyframe-residual mechanism significantly mitigates this issue; however, the outputs still exhibit noticeable blurring. Reliable decodings, indicative of proper initialization, are achieved exclusively by combining key frames and residuals with the conduction of regularization.

Upon further training of the diffusion model, employing a 3D VAE without both 2D VAE and regularization leads to significant inconsistencies in the latent distribution at the onset of training. This largely disrupts the pre-trained diffusion model's initial weights, resulting in poor convergence and subpar performance. Although incorporating a 2D VAE to compress key frames partially mitigates this issue, it still only yields suboptimal results. It is solely by utilizing our proposed DP-VAE that the diffusion transformer training can effectively integrate dynamic mechanisms into the pretrained model's image generation capabilities, thereby achieving optimal performance.

The quantitative results presented in Table 3 demonstrate that omitting a 2D VAE for key frame compression leads to significantly poor reconstruction and final generation performance. While introducing regularization partially mitigates this issue, the suboptimal reconstruction performance continues to limit the quality of the generated results. Incorporating a 2D VAE without regularization yields the best reconstruction performance. However, due to inconsistencies in the initial distribution, training the diffusion model still partially disrupts the original weights, resulting in suboptimal performance. In contrast, our proposed DP-VAE strikes an optimal balance between reconstruction performance and distribution preservation, resulting in superior generation performance.

**2D→3D vs. 2D+3D Attention Mechanism.** Previous methods (Blattmann et al., 2023b; Guo et al., 2024) propose to incorporate additional temporal layers and freeze original pre-trained spatial weights to preserve spatial features and enhance performance. Similarly, we can also introduce a trainable copy of the original spatial weights dedicated to global attention, while retaining the existing spatial attention layers. During training, the original spatial attention weights remain frozen and only the newly added global attention weights are updated. We refer to this integrated method as "2D+3D" attention. However, as shown in Table 4, the performance of "2D+3D" does not significantly differ from that of "2D→3D" (our default setting introduced in Section 3.2) and results in a considerably higher parameter count (4.15B vs. 2.03B). This demonstrates that our proposed "2D→3D" conversion is both simple and efficient, offering a promising approach to adapting pre-trained image Diffusion Transformers for video generation. Detailed implementations of the "2D+3D" conversion are provided in the supplementary material.

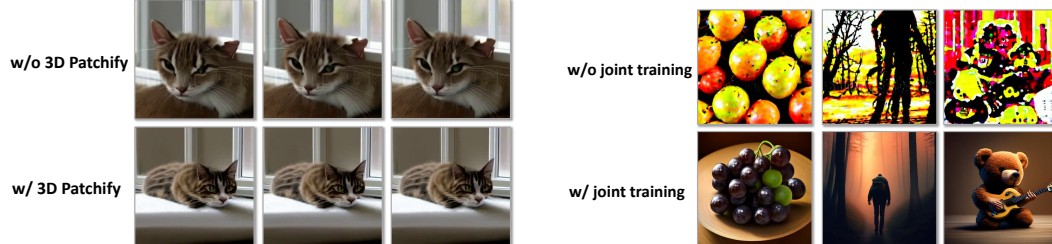

Figure 6: **Generated videos without and with 3D patchify.** The prompt is "A cat sleeping on a windowsill."

Figure 7: **Generated images without and with joint image-video training.**

|  | Param. | FVD (↓) | IS (↑) |
|---|---|---|---|
| 2Dand3D | 4.15B | 563.24 | 29.00 |
| 2Dto3D | 2.03B | 556.42 | 29.61 |

Table 4: Ablation of "2Dand3D" and "2Dto3D" variants.

|  | FVD (↓) | IS (↑) |
|---|---|---|
| 2D patchify | 756.25 | 27.85 |
| 3D patchify | 556.42 | 29.61 |

Table 5: Ablation study of 3D patchify.

|  | FVD (↓) | IS (↑) |
|---|---|---|
| w/o | 556.42 | 29.61 |
| w/ | 543.18 | 32.47 |

Table 6: Ablation of joint image-video training.

**3D Patchify.** Figure 6 illustrates that generated results without 3D patchify exhibit pronounced block artifacts. This issue arises because each token lacks awareness of its temporal position, hindering the accurate modeling of temporal causality. Quantitatively, as presented in Table 5, models without 3D patchify achieve significantly higher Frechet Video Distance (FVD) scores, indicating inferior performance.

**Joint Image-Video Training.** We adhere to the default settings outlined in Section 4.1, setting the probability of video generation training at 80%. Figure 7 illustrates that under the joint image-video training strategy, our model preserves the original text-to-image (T2I) capabilities. In contrast, without joint image-video training, the model's ability to generate images is entirely compromised. Table 6 demonstrates that joint image-video training not only maintains the T2I generation capability but also enhances video generation performance.

## 5 LIMITATION

We utilized the open-source SD3 medium (Esser et al., 2024) as the base model to build our VideoDiT model, which comprises approximately 2 billion parameters. Then the VideoDiT model is trained on a video dataset consisting of 1.3 million samples using 8 NVIDIA H100 GPUs (80GB). Due to limitations in computational and data resources, we are unable to fully explore the upper performance bounds of VideoDiT, despite its commendable performance under these constrained conditions. In future work, our aim is to investigate the performance limits of this approach using larger-scale models, increased computational capacity, and expanded data resources.

## 6 CONCLUSION

In this study, we introduce VideoDiT, a framework designed to extend the capabilities of large-scale pre-trained text-to-image (T2I) diffusion models to video generation by seamlessly integrating motion cues with minimal modifications. Specifically, our approach addresses adaptations in both the tokenizer and the diffusion model components. For the tokenizer, we propose the Distribution-Preserving VAE (DP-VAE), which compresses videos in both spatial and temporal dimensions while maintaining the original distribution. Regarding the diffusion model, we propose to directly convert existing spatial attention mechanisms to global attention, thereby incorporating a comprehensive receptive field without introducing additional parameters, thus providing an effective initialization strategy. Extensive experiments validate the effectiveness of our proposed DP-VAE and the adapted video diffusion transformer, demonstrating significant improvements in video reconstruction fidelity and generation quality.

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

# A  DETAILED IMPLEMENTATIONS

We strongly recommend downloading the **supplementary materials** provided. After extracting the contents, navigate to the designated folder and click on **"index.html"** to view the dynamic video results.

## A.1  DISTRIBUTION-PRESERVING VAE

For the key frame encoder $\mathcal{E}_k$ and decoder $\mathcal{D}_k$, we directly utilize the pre-trained models from Stable Diffusion 3 (Esser et al., 2024). The implementation of the 3D VAE encoder $\mathcal{E}_r$ and the decoder $\mathcal{D}$ adheres to the factorized pseudo-3D mechanism (Singer et al., 2022; Blattmann et al., 2023b). Specifically, we initialize a trainable copy of the pre-trained 2D VAE and incorporate corresponding 1D components working on temporal dimension after each 2D ResBlock and Attention Block, enabling the model to effectively process both spatial and temporal dimensions. Additionally, temporal downsampling and upsampling operations are performed following the first two spatial downsampling layers and the last two upsampling layers, respectively, resulting in a $4\times$ downsampling factor along the temporal dimension.

## A.2  UNIFYING IMAGE DIFFUSION TRANSFORMER TO VIDEO GENERATOR

**3D Patchify.** Figure 8 illustrates the detailed 3D patchify process designed to incorporate temporal positional information. We introduce an additional branch dedicated to integrating 3D positional embeddings and implement a zero-initialized convolutional projection layer to facilitate progressive updating.

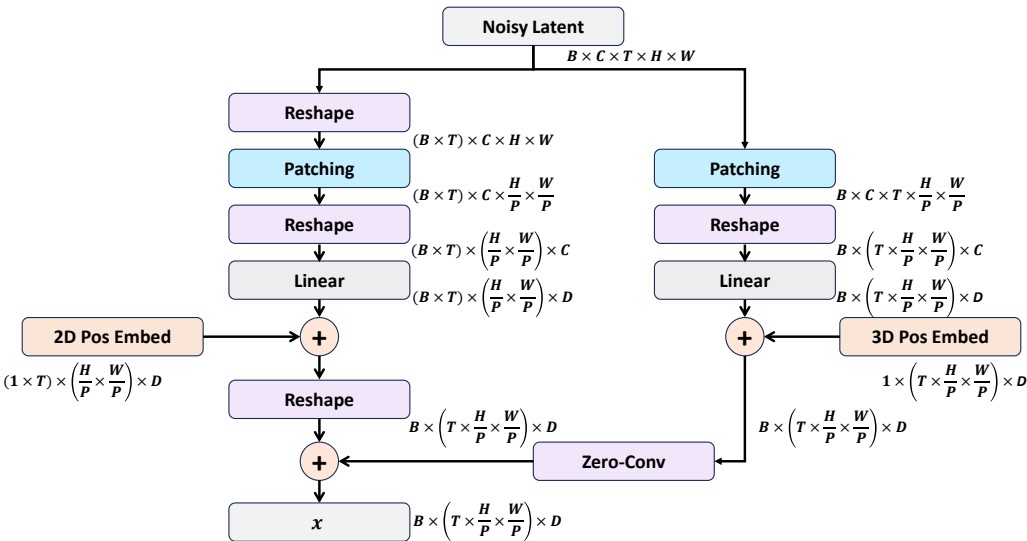

Figure 8: Our proposed 3D patchify process to incorporate temporal positional information.

**2D+3D Attention Mechanism.** Figure 9 illustrates the "2D+3D" attention variants described in Section 4.3. Specifically, we attach an additional global attention block after each of the original spatial attention blocks. The weights of these global attention blocks are initialized by copying the weights from their previous 2D spatial attention blocks. Furthermore, we incorporate zero-initialized convolution layers (Singer et al., 2022; Zhang et al., 2023b) and skip-connection to facilitate progressive updating. This initialization strategy ensures that the additional global attention blocks begin with weights equivalent to the original 2D attention blocks, thereby promoting seamless integration and effective adaptation within the model architecture.

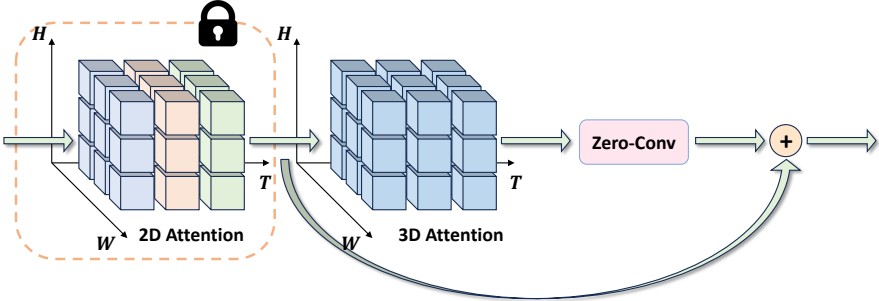

Figure 9: Illustration of the "2D+3D" attention variant in the Ablation Study.

**Joint Image-Video Training.** During training, we employ a probability parameter, $p$, to determine whether the current iteration is dedicated to video generation or image generation. If the iteration is designated for image generation, the input image is encoded using the original 2D VAE encoder, $\mathcal{E}_k$, to obtain the latent variable. Conversely, if the iteration is designated for video generation, the input video is encoded using the encoder of our proposed DP-VAE. In the forward propagation process, global attention mechanisms are applied no matter the input is an image or a video. During inference, images or videos are generated by first denoising sampled noise from a standard Gaussian distribution and then decoding using the corresponding decoder. Notably, no additional conditions are required to specify the type of the current iteration, since positional embeddings function as conditional inputs for both image and video data in each training stage.

**Rectified Flow.** We utilize the pre-trained Stable Diffusion 3 (SD3) (Esser et al., 2024) as the foundational text-to-image (T2I) diffusion model in our framework. The SD3 models employ Rectified Flows (Liu et al., 2023; Albergo & Vanden-Eijnden, 2023) to construct a transport mapping $T$ between a standard Gaussian distribution $\pi_0$ and the data distribution $\pi_1$, formulated as

$$X_t = (1 - t)X_0 + tX_1, \tag{5}$$

where $X_t$ represents the noisy input at time step $t$, $X_0$ is the sampled noise from $\pi_0$, and $X_1$ denotes the clean data from $\pi_1$. The neural network is trained to directly predict the velocity $v_\Theta = X_1 - X_0$.

## B    ADDITIONAL RESULTS

We present additional results of generated images of our proposed VideoDiT in Figure 10.

We present additional results of generated videos of our proposed VideoDiT in Figure 11.

We present additional reconstruction comparisons between our proposed DP-VAE and other methods in Figures 12 to 14.

We recommend downloading and extracting the provided files, then opening `index.html` to view the dynamic videos.

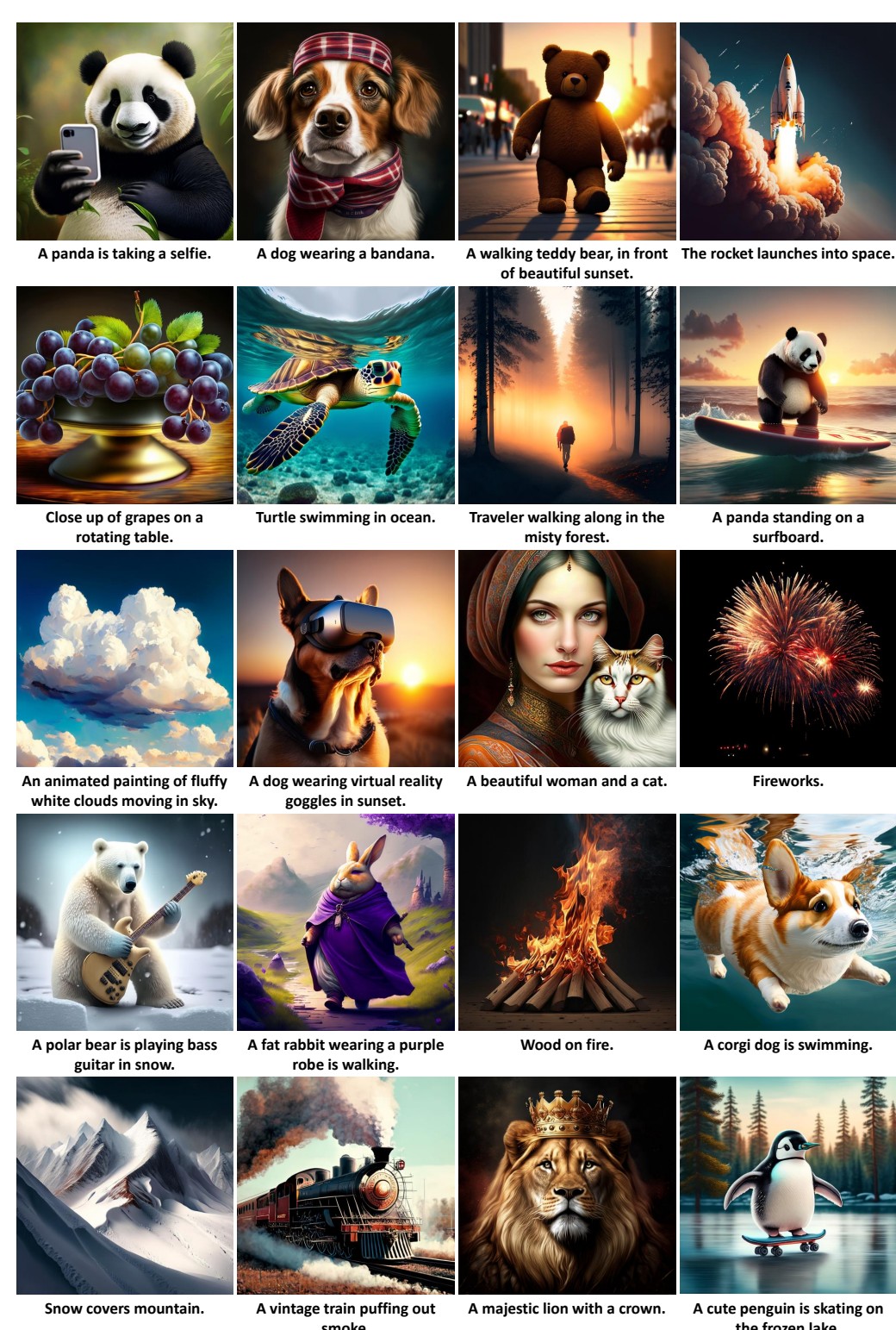

Figure 10: Generated images of VideoDiT.

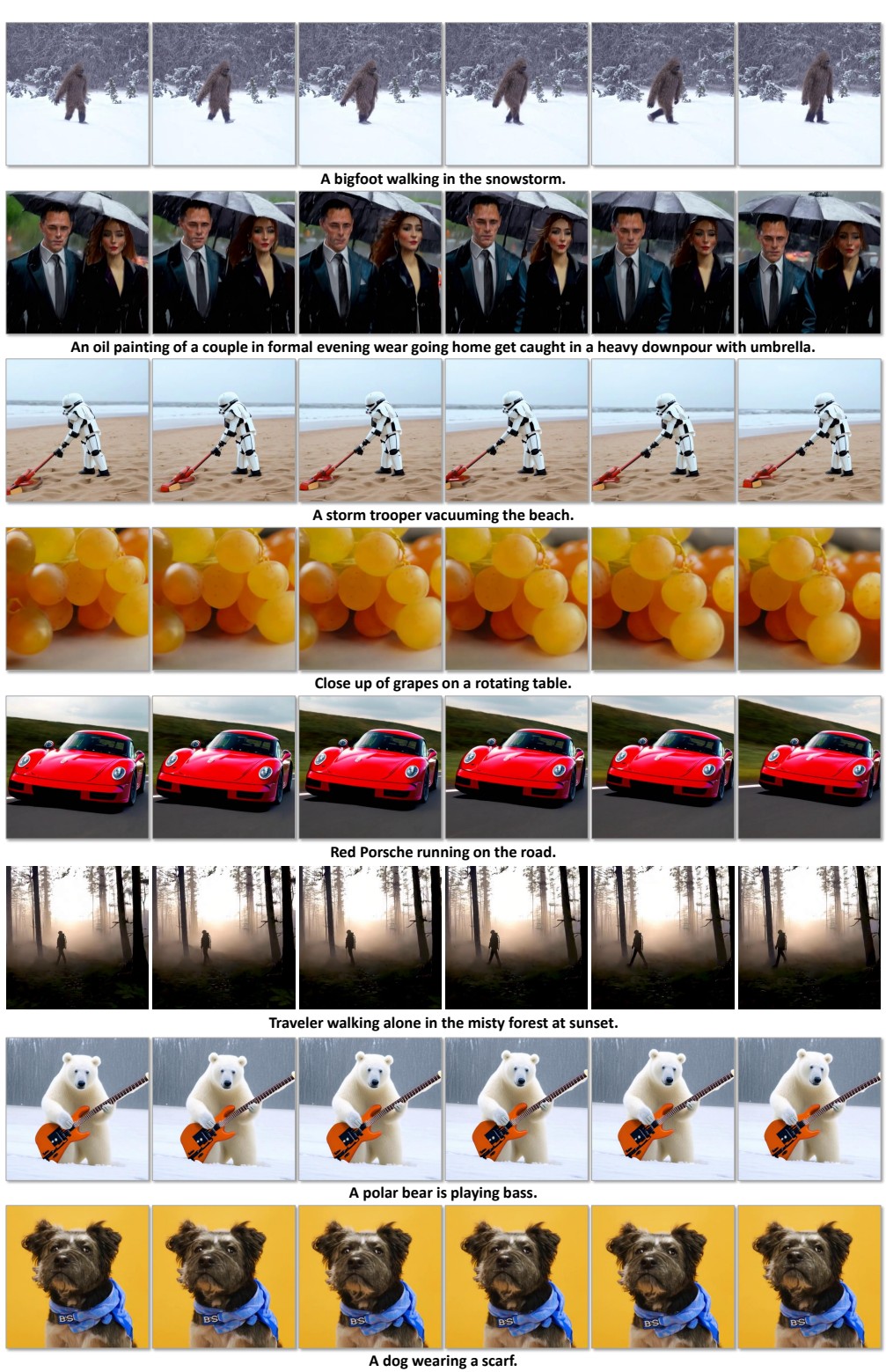

Figure 11: Generated videos of VideoDiT.

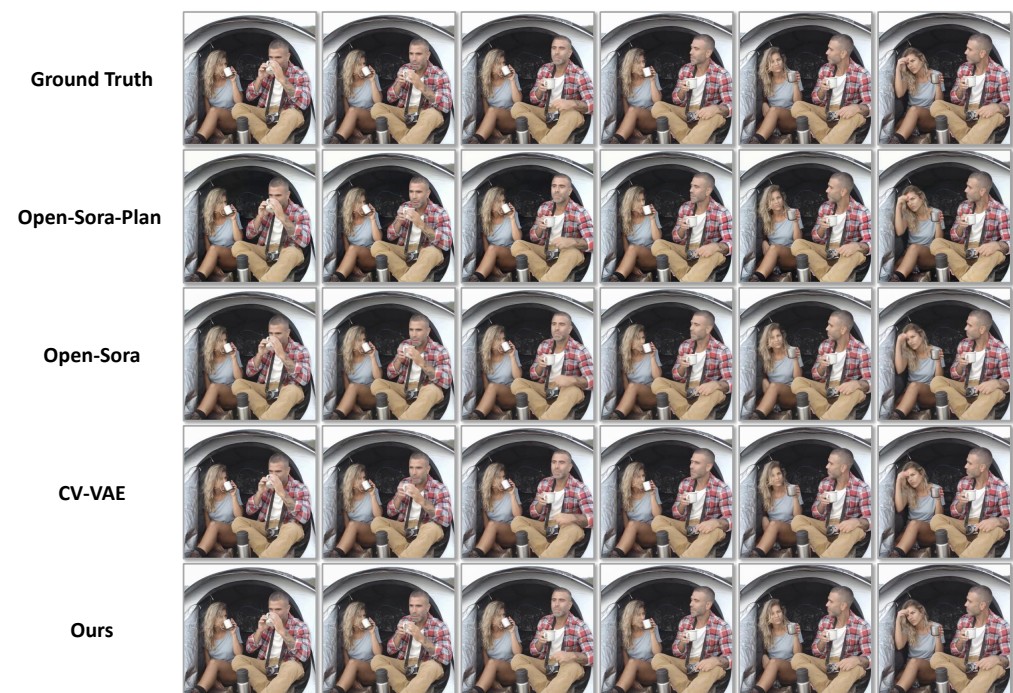

Figure 12: Additional qualitative comparison of our proposed DP-VAE and other methods. Best viewed with a zoomed-in view.

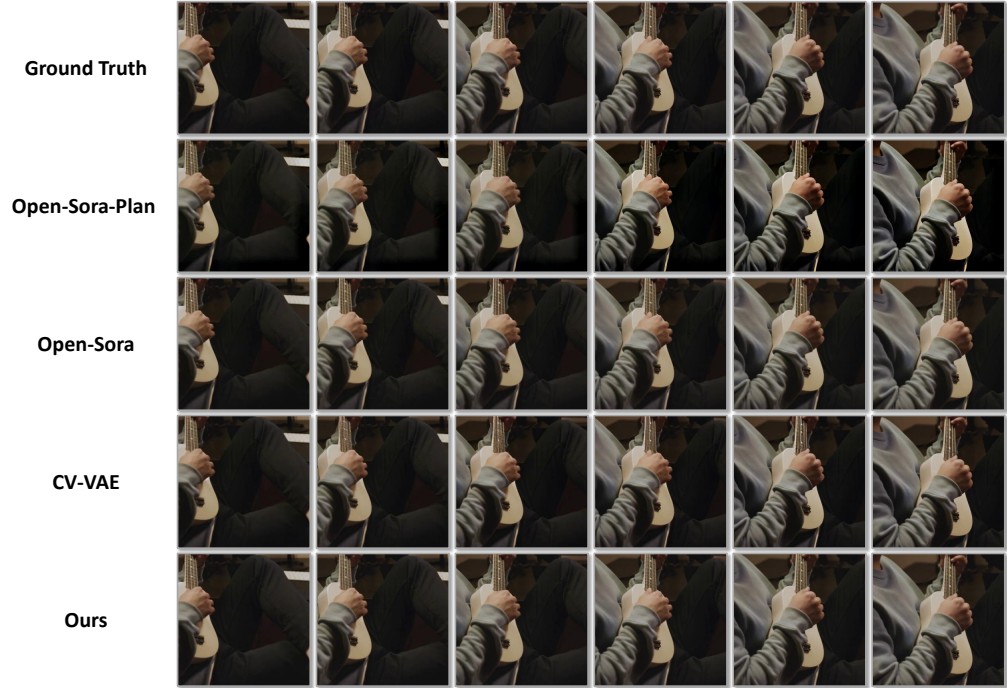

Figure 13: Additional qualitative comparison of our proposed DP-VAE and other methods. Best viewed with a zoomed-in view.

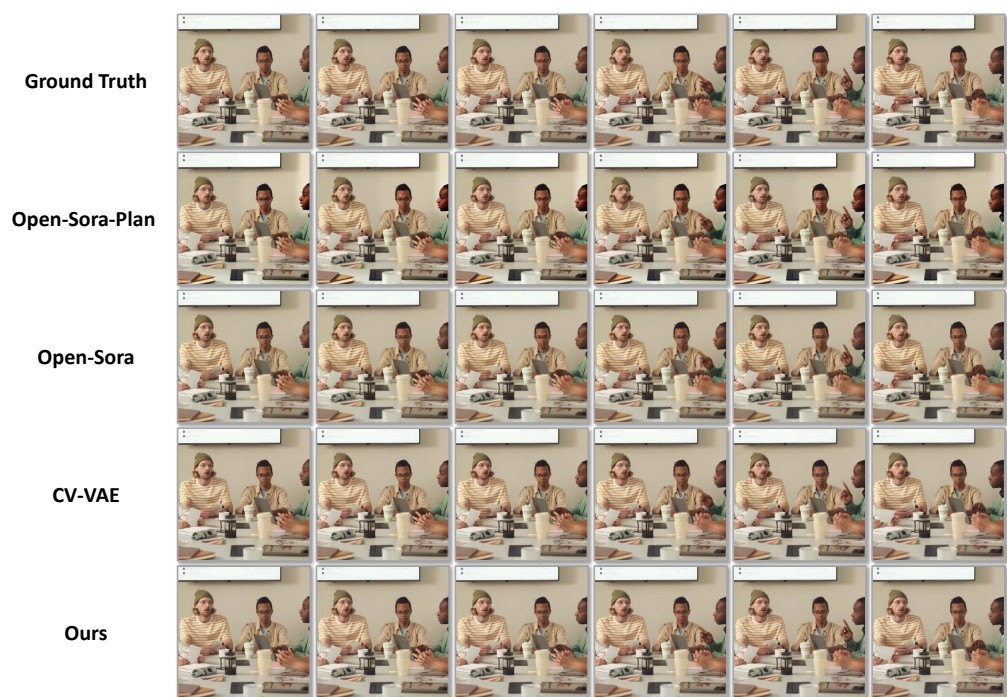

Figure 14: Additional qualitative comparison of our proposed DP-VAE and other methods. Best viewed with a zoomed-in view.

