# OpenReview forum: "VideoDiT: Bridging Image Diffusion Transformers for Streamlined Video Generation"
_ICLR.cc/2025/Conference — Submitted to ICLR 2025_

### Official Review · Reviewer_xV9d · 2024-10-24

**Soundness:** 2
**Presentation:** 2
**Contribution:** 2
**Rating:** 3
**Confidence:** 5

**Summary:**

In this paper, the authors propose a framework called VideoDiT for extending a text-to-image diffusion model into text-to-video model without much new learnable parameters. In particular, this work has two main contributions. One is called DP-VAE, which extends a 2D image VAE into a 3D video VAE. The main difference from previous video VAE is that it tries to align the latent distribution of video latent with image latent, so that we can seamlessly leverage the image pretrain on video training. The second contribution is on how to extend 2D DiT-based image diffusion model into 3D video modeling. Experiments are conducted on both VAE reconstruction evaluation and video generation quality comparison.

**Strengths:**

1. This paper focuses on an important research problem, which has potential benefit to the downstream applications. Since the image/video generation will all need a good video tokenizer.
2. The intuition of this work makes sense to me, where it's better to train a video generation model based on a pretrained image model, so that we can maximize the utilization of all existing research practices in building strong image generation models.
3. The experiments are conducted on both auto-encoding reconstruction and generation, which can reflect both the reconstruction quality and whether it can indeed boost the generation process, which is quite good.

**Weaknesses:**

1. The method illustration is quite unclear to me, where I'm confused by many parts of the approach. For example, how do we choose the key frame? Is the key frame always the first frame? What if there're several scene changes, so that the encoded image latent is largely different from the video latent. Why is the compression ratio 4? More questions are elaborated in the next section. Please provide a more detailed explanation or diagram of the key frame selection process and how it handles different scenarios like scene changes. Additionally, please clarify on how the compression ratio is determined and applied consistently across different video lengths.
2. I would suggest the authors to also include the inference speed (wall-clock time) and memory cost on encoding-decoding a video. Since the proposed framework actually has more FLOPs - one image VAE + one video VAE - I think the inference speed might be slow and memory cost would be higher compared to baselines. Please provide a table comparing inference speed and memory usage for the proposed method versus baselines on standardized video lengths and resolutions.
3. I would suggest the authors to highlight the compression ratio for baselines and the proposed video VAE. Since it's unclear to me whether all methods have the same compression ratio, hence whether this is a fair comparison. Please add a column in the comparison tables that explicitly states the compression ratio for each method, or to clarify in the text how you ensured comparable compression ratios across methods.
4. Novelty. The DP-VAE part is generally OK to me, conditioned on the authors have clarified my unclear points. However, the extension from image DiT to video DiT has no clear novelty to me. I think it's already a quite common practice for all the recent video DiT generators. They would even incorporate better position embedding extension like 2D RoPE -> 3D RoPE, so the section 3.2 is actually of no novelty to me. When changing from image to video under the DiT architecture, except for the position embedding change, it's always the full attention and I see nothing new.
5. The generation results still have motion blur and not that high-quality to me.
6. I'm wondering whether this method performs well on higher compression ratio. Since 4x8x8 compression ratio is still too low compared to current SOTA video generators (e.g., Meta MovieGen uses 8x8x8), the sequence length would be unaffordable long. Please include an experiment or discussion on how your method performs with higher compression ratios, specifically comparing to the 8x8x8 ratio.

**Questions:**

1. The authors mentioned that the video encoding is decomposed into key frame and residuals. What's the meaning of this? If I have a video of 10 frames and regard the first frame as key frame, how do we derive residual for the 2-10 frames? By substracting each frame with the first frame? If the authors just input x_1 to image VAE and x_2, ..., x_10 to the video VAE, then why do the authors call it as residual? Is it simply because the encoded latent is added to the image latent?
2. How is the key frame decided at the training and inference stage? Is it always the first frame? If not, does this mean we have to additionally choose the keyframe before auto-encoding.
3. I'm still quite confused by the compression ratio of the proposed DP-VAE. The authors claim to have temporal compression ratio of 4, but this is without the considering of key frame. For example, if I have 13-frame video, and the first frame is regarded as keyframe. The actual compression ratio is 1+ (13 -1 ) // 4 = 4, which is less than 4, and heavily depends on the encoded video sequence length.

---

### Official Review · Reviewer_JqLE · 2024-10-29

**Soundness:** 2
**Presentation:** 3
**Contribution:** 2
**Rating:** 3
**Confidence:** 5

**Summary:**

This paper presents a method for video generation by adapting the pre-trained T2I models. Mainly, it proposes an innovation scheme termed Distribution-Preserving VAE (DP-VAE), which leverages both the 3D VAE and 2D VAE based on the keyframe selections. It also introduces the regularization loss to better align the 3D and 2D latent distributions. This modified VAE was trained on a self-collected video dataset. Other than that, the paper also explores the connection between 2D and 3D attention for better initialization.

**Strengths:**

The main strength of this work is the proposed Distribution-Preserving VAE (DP-VAE). It provides a solution to transform a 2D VAE into a 3D VAE. This is achieved by determining the keyframe in the input video.  The proposed DP-VAE also introduces a regularization loss in the reconstruction of 2D keyframes and the alignment of latent codes in 3D VAE. The proposed solution looks simple and easy to implement.

The paper is easy to read, and the methods are well described, except some details are missing. The clarity of the paper in terms of presentation quality should be good. The paper does offer certain benefits in guiding how to effectively utilizing pre-trained image diffusion models.

**Weaknesses:**

The proposed DP-VAE offers an interesting solution on how to adapt the 2D VAE to the 3D VAE. However, the contributions are limited, and many details need to be clarified.

Regarding the DP-VAE part, it has many details that need to be addressed, which is also the biggest concern so far. First, the paper does not clarify how the keyframes were selected. Merely it mentions the Groups of Pictures (GoPs) from the video compression. However, it will be necessary to specify how the key video frames were selected in this task. For instance, what are the algorithms or criteria used to perform the keyframe selection? This selection significantly affects the DP-VAE design.

Secondly, the architecture presented in Figure 1, as well as the descriptions presented, contains some confusion. It has been mentioned that the 3D VAE has been applied to the entire video segment x in Section 3.1. But in the description of some other places (see Abstract description), the 3D VAE was employed to deal with the non-keyframe compression.

Thirdly, the DP-VAE was trained on a self-collected video dataset, which raises concerns for fair comparisons and reproducibility. Could you discuss how to ensure the fair comparisons given the use of a custom dataset? More details regarding the self-collected dataset and its usage purpose would be appreciated.

Finally, the results presented in Table 3 offer some contradicting observations regarding the effectiveness of 2D VAE and regularization, which are not very well supported with explanations. The results show that the proposed method only offers a trade-off, other than providing the best performance. The paper mentions, "Incorporating a 2D VAE without regularization yields the best reconstruction performance". But this training will result in original weight disruption, leading to suboptimal performance. Please also provide more details in this aspect.

The regularization term of DP-VAE puts constraints on the key-frame reconstruction and the 3D latent variables. This is a commonly used regularization in the VAE design but applied to different sets of inputs here. Therefore, the contribution is not significant.

The paper also explores the design of transferring 2D attention to 3D attention. It claims the direct copy of 2D attention for 3D attention serves as an effective initialization strategy without introducing additional parameters in the method section. However, in the experimental section, the paper refers the a trainable copy of the original spatial weights dedicated to global attention as the “2D+3D” attention, which differs from the 2D→3D. The two schemes need to be further clarified with more details.

Other parts such as Joint Image-Video Training have been explored in the existing video generation works such as VideoCrafter, Lavie and Open-Sora.

The paper shows the results in Table 2 and claims that the proposed approach uses a small amount of video training data, but still achieves competitive performances. While this claim makes sense, the reviewers would also be interested to know if the same set of training data was used, will the performance further surpass the existing ones? The paper also uses self-collected videos, it could be a concern for fair comparisons.

**Questions:**

The DP-VAE needs to be clarified, especially the choice of keyframe selections. The architecture regarding the DP-VAE also needs to be clarified on whether the 3D VAE is applied to the entire segment or just the non keyframe. The paper also needs to address why it adopts self-collected datasets, other than using the public ones.

---

### Official Review · Reviewer_TuNq · 2024-11-01

**Soundness:** 1
**Presentation:** 2
**Contribution:** 1
**Rating:** 1
**Confidence:** 5

**Summary:**

The paper presents VideoDiT, a DiT framework for video generation. The main contributions are the following:
1) DP-VAE, auto-encoder that is initialized from image encoder and fine-tuned for videos to preserve the original latent distribution.
2) Some modifications to DiT, such as global attention and 3D patchify and joint image-to-video training.

**Strengths:**

- The paper is fairly well written and easy to understand.

**Weaknesses:**

In the light of recent development in the community, it is clear that the amount of resources that can be used for training of the video model, is proportional to the final quality of these models. Having said this, there is an avenue for two types of papers: (1) papers that try to push generation quality to the limit at a large scale, and (2) papers that present some novel insight or technique on a small scale and thoroughly evaluate it.

This particular paper is in-between and neither demonstrates a good quality of generations, nor presents any novel insight or technique. Moreover it does not provide thoughtful evaluation of the proposed techniques.

In more details:
- The quality of the results provided in supplementary material is not very good. It can be even said that the model did not learn anything, since the motion is meaningless and the quality of individual images is just inherited from the pre-trained SD3 model. So if we evaluate this model on (1), models like [1, 2, 3] which are also diffusion transformers demonstrate much better results.

- The modification proposed by this paper was already proposed in other papers. Initializing the auto-encoder from image one was done in [5]. Video DiT was used in [2, 4], the modification to Video DiT provided in this paper is minimal.

- The evaluation of VAE does not include the account number of channels used. DP-VAE has 16 channels, while others have 4. It is clear that VAE with more channels will have better reconstruction. So a fair comparison will be against VAE of [1] which also has 16 channels.

- The main model is evaluated at zero-shot UCF-101, which is highly dependent on the choice of the training videos and overall is not a reliable benchmark. The better approach would be to use some video evaluation benchmark such as VBench[6] or VideoScore[7], or do user study.

[1] CogVideoX: Text-to-Video Diffusion Models with An Expert Transformer

[2] Photorealistic Video Generation with Diffusion Models

[3] Snap Video: Scaled Spatiotemporal Transformers for Text-to-Video Synthesis

[4] Latte: Latent Diffusion Transformer for Video Generation

[5] Cv-vae: A compatible video vae for latent generative video models.

[6] VBench : Comprehensive Benchmark Suite for Video Generative Models

[7] VideoScore: Building Automatic Metrics to Simulate Fine-grained Human Feedback for Video Generation

**Questions:**

--

**Details Of Ethics Concerns:**

--

---

### Official Review · Reviewer_ZP15 · 2024-11-03

**Soundness:** 2
**Presentation:** 2
**Contribution:** 2
**Rating:** 3
**Confidence:** 4

**Summary:**

Key Contributions:

1. DP-VAE (3D VAE for Video Compression):

- Residual Model: Compresses additional information into key-frame latents derived from a 2D VAE.

- Regularization Loss: Aligns the latent spaces of 3D and 2D VAEs to enhance compression efficiency.

2. Video-DiT Diffusion Model:

- Operates in the latent space of DP-VAE.

- Transitions from T2I MMDiT to T2V by extending the 2D patchify operation to 3D and modifying 2D attention mechanisms to 3D attention.

Despite these innovative concepts, the manuscript requires significant revisions for acceptance at the conference. Please refer to the Weaknesses and Questions sections for detailed critiques.

**Strengths:**

1. DP-VAE: The idea of DP-VAE seems novel and has can have a good potential.

2. Clearness: The method is simple and easy to understand.

**Weaknesses:**

1. Video reconstruction results: The reconstruction performance in Table 1 is reported on a small subset of 50 videos randomly selected from the Pexels dataset. This benchmark for evaluating video reconstruction performance is novel and, as such, has not been previously reported in the literature. However, it contains too few samples to be considered robust. A more meaningful evaluation benchmark should include at least 1,000 videos. It is suggested to evaluate models on a larger dataset, for example WebVid eval [4] set as in CV-VAE paper [1].

2. Missed VAE comparisons: Table 1 does not include comparisons with CogVideoX VAE [2] and Open-Sora VAE [3], which are significant baselines in this field.

3. Video quality metrics: Table 1 reports only frame-wise video quality metrics and neglects any temporal consistency metrics. It is recommended to measure the Fréchet Video Distance (FVD) between original and reconstructed videos (rFVD) to provide a more comprehensive evaluation.

4. Video generation results: The results in Table 2 make it difficult to assess the potential of the proposed video generation method. While VideoDiT shows comparable performance when trained on a smaller dataset, it remains unclear if the model would outperform competitors when trained on larger datasets. For the purity of the experiment it is recommended to train VideoDiT on the full WebVid-10m [4] dataset and compare its performance to baselines trained on the same data.

5. Zero-shot T2V benchmark: The evaluation benchmark used in Table 2 is known to have a poor correlation with human perception [6]. It is recommended to use more sophisticated benchmarks for evaluation, such as Vbench [7], to provide a more comprehensive assessment of the model's performance.

6. Missed T2V comparisons: Table 2 does not include comparisons with CogVideoX [2], Open-Sora [3], and Open-Sora-Plan [3], which are significant baselines of Video DiT architectures.

**Questions:**

My main concerns and comments regarding the paper are related to the experiment section:

1. Video reconstruction results in Table 1 would benefit from including results for CogVideoX VAE [1] and Open-Sora VAE [3], adding rFVD metric, and evaluation on a more reliable dataset (e.g. WebVid eval [4]). Please refer to weaknesses 1, 2, and 3 for more details.

2. I suggest the authors to revise the video generation experiments by evaluating Vbench [7] benchmark, including comparison with CogVideoX [2], Open-Sora [3], and Open-Sora-Plan [3], and comparing with baselines in the same setup - fixed dataset and computation amount. Please refer to weaknesses 4, 5, and 6 for more details.



The current quality of the experimental results prevents me from giving a favorable recommendation for this article. To change my judgment, the experiments would need to be substantially redesigned.







[1] Zhao, S., Zhang, Y., Cun, X., Yang, S., Niu, M., Li, X., ... & Shan, Y. (2024). CV-VAE: A Compatible Video VAE for Latent Generative Video Models. arXiv preprint arXiv:2405.20279.

[2] Yang, Z., Teng, J., Zheng, W., Ding, M., Huang, S., Xu, J., ... & Tang, J. (2024). CogVideoX: Text-to-video diffusion models with an expert transformer. arXiv preprint arXiv:2408.06072.

[3] Zangwei Zheng, Xiangyu Peng, Tianji Yang, Chenhui Shen, Shenggui Li, Hongxin Liu, Yukun Zhou, Tianyi Li, and Yang You. Open-sora: Democratizing efficient video production for all, March 2024. URL https://github.com/hpcaitech/Open-Sora.

[4] Bain, M., Nagrani, A., Varol, G., & Zisserman, A. (2021). Frozen in time: A joint video and image encoder for end-to-end retrieval. In Proceedings of the IEEE/CVF international conference on computer vision (pp. 1728-1738).

[5] Chen, T. S., Siarohin, A., Menapace, W., Deyneka, E., Chao, H. W., Jeon, B. E., ... & Tulyakov, S. (2024). Panda-70m: Captioning 70m videos with multiple cross-modality teachers. In Proceedings of the IEEE/CVF Conference on Computer Vision and Pattern Recognition (pp. 13320-13331).

[6] Girdhar, R., Singh, M., Brown, A., Duval, Q., Azadi, S., Rambhatla, S. S., ... & Misra, I. (2023). Emu video: Factorizing text-to-video generation by explicit image conditioning. arXiv preprint arXiv:2311.10709.

[7] Huang, Z., He, Y., Yu, J., Zhang, F., Si, C., Jiang, Y., ... & Liu, Z. (2024). Vbench: Comprehensive benchmark suite for video generative models. In Proceedings of the IEEE/CVF Conference on Computer Vision and Pattern Recognition (pp. 21807-21818).

[8] PKU-Yuan Lab and Tuzhan AI etc. Open-sora-plan, April 2024. URL https://doi.org/10. 5281/zenodo.10948109.

---

### Meta-Review · Area_Chair_H16L · 2024-12-16

**Metareview:**

Summary: This paper proposes a Distribution-Preserving VAE (DP VAE) framework, which processes video frames by encoding the first frame using a pre-existing 2D VAE and applying a 3D VAE to the subsequent frames. A regularizing loss is introduced to align the latents. Additionally, the authors present VideoDiT, which extends the DP-VAE latents by incorporating 3D attention over spatio-temporal latents, adapted from 2D attention.
Strengths: The idea of separating I- and P-latents in a VAE framework is conceptually relevant as it leverages redundancies in video data, unlike traditional 3D VAEs that treat all frames uniformly.
Weaknesses: The motivation for selecting the first raw frame as the I-frame is insufficiently explained. The paper  lacks comparisons with well-established Video VAE baselines, which significantly undermines its contribution. The VideoDiT contribution, while extending 2D models to videos, is a straightforward and already common practice. Relevant citations provided by reviewers highlight similar approaches. The results are not particularly strong and do not convincingly support the proposed methods.
Rejection Reason: The paper falls short due to the lack of depth and rigor in its problem formulation and proposed solutions. It presents straightforward contributions that are not adequately motivated and misses critical comparisons with prior work.

**Additional Comments On Reviewer Discussion:**

Overall, the paper does not meet the acceptance threshold. While the approach of using a 2D VAE for keyframes and a 3D VAE for remaining frames is an interesting direction, the work suffers from a lack of rigor in motivation and problem formulation. The solutions are straightforward and not novel. The paper received 3x Reject and 1x Strong Reject ratings. Authors did not submit a rebuttal, and the ratings remained unchanged.

---

### Decision · Program_Chairs · 2025-01-22

Reject